# Evolving Therapeutic Approaches for Older Patients with Acute Myeloid Leukemia in 2021

**DOI:** 10.3390/cancers13205075

**Published:** 2021-10-11

**Authors:** Irene Urbino, Carolina Secreto, Matteo Olivi, Vincenzo Apolito, Stefano D’Ardia, Chiara Frairia, Valentina Giai, Semra Aydin, Roberto Freilone, Chiara Dellacasa, Luisa Giaccone, Dario Ferrero, Ernesta Audisio, Alessandro Busca, Marco Cerrano

**Affiliations:** 1Department of Oncology, Division of Hematology, Presidio Molinette, AOU Città della Salute e della Scienza di Torino, 10126 Turin, Italy; iurbino@cittadellasalute.to.it (I.U.); csecreto@cittadellasalute.to.it (C.S.); molivi@cittadellasalute.to.it (M.O.); vapolito@cittadellasalute.to.it (V.A.); sdardia@cittadellasalute.to.it (S.D.); cfrairia@cittadellasalute.to.it (C.F.); vgiai@cittadellasalute.to.it (V.G.); rofreilone@cittadellasalute.to.it (R.F.); eaudisio@cittadellasalute.to.it (E.A.); 2Department of Molecular Biotechnology and Health Sciences, Division of Hematology, University of Torino, 10126 Turin, Italy; lgiaccone@cittadellasalute.to.it (L.G.); dferrero@cittadellasalute.to.it (D.F.); 3Department of Oncology, Hematology, Immuno-Oncology and Rheumatology, University of Bonn, 53115 Bonn, Germany; semra.aydin@ukbonn.de; 4Department of Oncology, SSD Trapianto Allogenico, AOU Città della Salute e della Scienza di Torino, 10126 Turin, Italy; cdellacasa@cittadellasalute.to.it (C.D.); abusca@cittadellasalute.to.it (A.B.)

**Keywords:** acute myeloid leukemia, elderly, fitness, precision medicine, targeted therapy

## Abstract

**Simple Summary:**

The better understanding of disease biology, the availability of new effective drugs and the increased awareness of patients’ heterogeneity in terms of fitness and personal expectations has made the current treatment paradigm of AML in the elderly very challenging. Here, we discuss the evolving criteria used to define eligibility for induction chemotherapy and transplantation, the introduction of new agents in the treatment of patients with very different clinical conditions, the implications of precision medicine and the importance of quality of life and supportive care, proposing a simplified algorithm that we follow in 2021.

**Abstract:**

Acute myeloid leukemia (AML) in older patients is characterized by unfavorable prognosis due to adverse disease features and a high rate of treatment-related complications. Classical therapeutic options range from intensive chemotherapy in fit patients, potentially followed by allogeneic hematopoietic cell transplantation (allo-HCT), to hypomethylating agents or palliative care alone for unfit/frail ones. In the era of precision medicine, the treatment paradigm of AML is rapidly changing. On the one hand, a plethora of new targeted drugs with good tolerability profiles are becoming available, offering the possibility to achieve a prolonged remission to many patients not otherwise eligible for more intensive therapies. On the other hand, better tools to assess patients’ fitness and improvements in the selection and management of those undergoing allo-HCT will hopefully reduce treatment-related mortality and complications. Importantly, a detailed genetic characterization of AML has become of paramount importance to choose the best therapeutic option in both intensively treated and unfit patients. Finally, improving supportive care and quality of life is of major importance in this age group, especially for the minority of patients that are still candidates for palliative care because of very poor clinical conditions or unwillingness to receive active treatments. In the present review, we discuss the evolving approaches in the treatment of older AML patients, which is becoming increasingly challenging following the advent of new effective drugs for a very heterogeneous and complex population.

## 1. Introduction

Acute myeloid leukemia (AML) is a complex disease with heterogeneous clinical and biological features [1], which mostly occur in older adults. As highlighted by the Surveillance, Epidemiology, and End Results (SEER) database, the incidence rate progressively increases and more than 30% of AML patients are 75 years or older at time of diagnosis [2]. In recent years, new therapeutic options and improvements in supportive care led to a slight increase in life expectancy in the younger AML population (5-year overall survival (OS) of 60% for patients <50 years old), but the prognosis remains poor in older patients (5-year OS of <20% and <5% among patients aged 65–74 and >75, respectively).

Several patient- and disease-related factors contribute to the unfavorable outcomes observed in this age group. Not only do older AML patients have more comorbidities and poorer performance status at diagnosis, but the disease itself is characterized by a higher incidence of adverse risk cytogenetic abnormalities (e.g., monosomal or complex karyotypes) and molecular alterations, with several cases being secondary or therapy-related [3].

Therapeutic results with intensive chemotherapy have been globally unfavorable, with reduced tolerance to treatment and increased risk of early death, lower complete remission (CR) rates and limited long-term survival [3,4]. However, a subgroup of patients can benefit from intensive approaches, but no simple algorithm or age limit exits to identify them, and patients with the same age can in fact present extremely different clinical conditions.

The advent of a plethora of new drugs, both for patients potentially fit for intensive chemotherapy and for frailer ones, makes therapeutic decisions even more challenging. Additionally, the possibility of consolidation with allogeneic hematopoietic cell transplantation (allo-HCT) in this age group may also be considered as a potentially curative approach in a consistent group of patients, which nonetheless should be carefully selected.

In this review, we will discuss the evolving approaches in the treatment of older AML patients (excluding acute promyelocytic leukemia), focusing on the eligibility criteria for intensive chemotherapy and allo-HCT, the role of new drugs in different clinical contexts, the implications of precision medicine and the importance of quality of life, which should never be forgotten, especially in this age group. A detailed discussion of the treatment approaches in relapsed cases goes beyond the scope of this review and has been recently discussed elsewhere [5,6,7].

## 2. Fitness Assessment

### 2.1. FIT, UNFIT, FRAIL

The limited possibilities of long-term survival and the relatively high frequency of severe complications have historically led many hematologists to be reluctant to start antileukemic treatment in older AML patients. Indeed, relatively recent SEER data (2000 to 2009) showed that 60 % of AML patients ≥65 years of age remained untreated [8]. On the other hand, patients receiving active therapy have consistently shown a survival advantage compared to untreated ones [9], and this is even more true today with new drugs allowing more effective non-intensive approaches [10]. However, toxicities can be relevant, and a proportion of frailer and very old AML patients might still benefit from best supportive care only [11]. In this rapidly changing therapeutic scenario, specific guidelines helping clinicians to assess the frailty of elderly patients are lacking. Practically, it could be useful to divide patients in FIT (eligible for intensive chemotherapy potentially followed by allogeneic stem cell transplantation), UNFIT (candidates for less intensive antileukemic therapy) and FRAIL (who benefit more from palliative care alone) [12,13,14], although in the current therapeutic scenario this paradigm might not be adequate any longer, as discussed below.

The decision to start a treatment in an older patient with AML and the choice among the available options are often based on the evaluation of the treating hematologist, usually followed by discussion in an interdisciplinary board. However, even if the expertise of the clinician is important, the usefulness of a more objective assessment to identify blurrier age-related problems in different health domains has been proved [15,16,17]. In a retrospective monocentric study by Aydin and colleagues, the authors evaluated three scores for the assessment of patients’ fitness at diagnosis and compared them to a physician’s subjective evaluation. Both the physician’s assessment and the frailty scores were able to discriminate FIT from UNFIT patients in terms of overall survival. However, the agreement between the fitness scores and physician’s evaluation was only low/moderate, suggesting that objective scores may reveal problems not routinely detected by clinicians and supporting their role in making treatment decision [18].

Many scoring systems have been proposed to identify patients who can benefit from intensive treatments, but none of them are universally accepted and therefore recommended in AML guidelines [19,20]. Below, we analyze several scores potentially useful for the assessment of older AML patients’ fitness, which are summarized in Table 1.

### 2.2. Early Mortality and Survival Predictive Models

Several studies aimed to identify patients unlikely to benefit from intensive chemotherapy, analyzing disease-related factors and simple clinical and laboratory parameters.

Performing an analysis on 2483 patients with AML aged more than 60 years, Wheatley et al. [21] proposed a prognostic index of survival based on age, performance status, cytogenetic risk and AML type (de novo vs. secondary). The authors identified three risk groups with good, standard and poor prognosis and the index was validated both in intensively and non-intensively treated patients [21]. While this score could be certainly used to foresee the probability of survival in each patient, it does not help to define the best treatment approach.

Walter et al. [22] analyzed 3365 adults (also including younger patients) with newly diagnosed AML treated with intensive chemotherapy, and developed a model able to predict the risk of early death, an important parameter to guide treatment decision. Their multiparametric calculator, which included age, performance status, platelet count, white blood cells, circulating blast percentage, albumin level, creatinine and diagnosis of secondary AML, was able to predict early induction death for patients of all ages and was more accurate in predicting treatment-related mortality (TRM) than performance status or age alone. In addition, the authors showed that the predictive accuracy of such a model was only minimally affected by the elimination of age, suggesting that this parameter should not be used alone for the assignment of curative intensive treatments [22]. Kantarjian et al. [23] analyzed 446 AML patients aged more than 70 years treated with cytarabine-based intensive chemotherapy, with the aim to identify predictive factors of high induction (8-week) mortality. Age above 80 years, complex karyotypes, performance status ≥ 2 and elevated creatinine were the significant factors identified by multivariate analysis [23]. Krug et al., examining a cohort of 1406 intensively treated AML patients ≥ 60 years old, developed a web-based application which took into account age, body temperature, hemoglobin, platelet count, fibrinogen, LDH and diagnosis of secondary AML. The “AML score” obtained was shown to predict the probability of CR and the risk of early death after intensive induction chemotherapy, and the results were validated in an independent cohort [24].

All these models were originally intended to predict survival or therapy-related mortality in patients treated with intensive chemotherapy. Therefore, as pointed out by Palmieri et al., the use of these scores for fitness assessment is questionable, since survival can only be considered a “surrogate” of fitness [25]. Indeed, their aim is to anticipate the outcome once the treatment choice has already been made, rather than helping the clinician to choose the more suitable therapy; in addition, they are often therapy specific and thus may not be applicable to patients receiving different therapeutic regimens.

Following a slightly different approach, Malfuson et al. [26] analyzed the factors associated with poor survival in a cohort of 416 older AML patients, with the aim to identify decisional tools able to guide treatment choices. They showed that cytogenetics was the strongest independent decisional factor, while three other parameters (age, performance status and white blood cell count) had to be combined to reach a good specificity. Consequently, they proposed a decisional index based on these factors, suggesting that older patients with unfavorable cytogenetics or with at least two among age ≥75 years, performance status ≥2 and leukocytosis should not be treated with intensive chemotherapy [26].

### 2.3. Geriatric Assessment 

A comprehensive geriatric assessment (CGA) is a multidisciplinary diagnostic process able to detect medical, psychosocial and functional problems not otherwise identified by routine evaluation. It provides an objective and detailed evaluation of patients’ health status, which can help physicians to identify areas of vulnerability, to predict survival and toxicities, to assist in treatment decisions and to guide interventions. Therefore, it is recommended in cancer patients by the International Society of Geriatric Oncology and the National Comprehensive Cancer Network (NCCN) [27]. However, CGA is often perceived as time and resource consuming in the onco-hematological setting, since it requires a multidisciplinary team and complex procedures which go beyond routine clinical practice. Moreover, its efficacy is limited, unless followed by interventions, follow up and adaptation of care planning. In order to overcome these issues, easier geriatric assessment (GA) tools, not requiring a geriatrician evaluation but applicable by oncologists and hematologists, have been developed to evaluate fitness in elderly patients with cancer [28,29]. The G-8 score, developed and validated by Soubeyran et al. in 2008, included seven Mini Nutritional Assessment items (appetite, weight loss, motricity, body mass index, cognition and depression, self-related health, medications) and age, and patients scoring <14 on a 0–17 point scale were identified as those who should be referred to a geriatric unit for GA [30]. The G-8 score was then validated and improved in the cancer population [18,31,32]. Deschler et al. [33] analyzed the ability of several geriatric variables to predict the outcome of older patients with myelodysplastic syndrome (MDS) or AML: the Karnofsky score, the assessment of activity of daily life and fatigue showed a prognostic strength similar to disease-related factors such as poor risk cytogenetics and blast count. Therefore, the authors supported the use of geriatric and quality of life (QoL) assessment tools in clinical practice [33]. The same conclusion was drawn by Sherman et al., in a study focusing on the variables included in the GA in 101 newly diagnosed AML patients aged 65 years or older. Baseline comorbidity score, difficulty with strenuous activity and pain emerged as independent predictors of higher risk of death in a multivariate model including cytogenetic risk [34]. Similarly, Klepin et al. demonstrated that GA was predictive of survival in AML patients aged more than 60 years receiving intensive chemotherapy; OS was significantly shorter in patients showing impairment in cognition and objectively measured physical function [35].

### 2.4. Comorbidity and Organ Function Scores 

In cancer patients, multiple studies have demonstrated the impact of comorbidities on prognosis [36]. One of the most used comorbidity-based scores is the hematopoietic cell transplantation-specific comorbidity index (HCT-CI), which was developed in 2005 by Sorror and colleagues from the original Charlson Comorbidity Index (CCI) to predict non-relapse mortality (NRM) and survival in patients undergoing allo-HCT [37]. In 2014, the same group proposed a modified composite score integrating comorbidities and age, which outperformed age alone in predicting NRM and survival [38]. Although originally developed and routinely used to assess eligibility to allo-HCT, the Sorror score might also be applied to assess cancer patients’ fitness. Recently, Sorror et al. demonstrated that a composite model, combining age, augmented HCT-CI (with the addition of hypoalbuminemia, thrombocytopenia and high lactate dehydrogenase level) and cytogenetic/molecular risks, was able to estimate 1-year mortality after initial therapy in AML patients [39]. Importantly, while patients with low and intermediate scores showed a significant survival benefit when treated with intensive approaches compared to non-intensive ones, this difference was not significant in patients with a high score. Thus, the AML composite model could be used to identify patients who do not benefit from intensive chemotherapy, but further validation is needed [9].

In 2013, Ferrara and colleagues proposed another model based on critical organ comorbidities, to provide uniform and easily applicable criteria for the definition of fitness. An analytic hierarchy consensus process was used with the auspices of the Italian Society of Hematology (SIE), Italian Society of Experimental Hematology (SIES) and Italian Group for Bone Marrow Transplantation (GITMO). The authors proposed a list of conceptual and operational criteria to define unfitness to intensive and non-intensive chemotherapy in AML patients, thus classifying them FIT, UNFIT or FRAIL. The criteria included age, performance status and several comorbidities, namely cardiac, pulmonary, renal, hepatic, infections, mental illness and uncontrolled neoplasia [40]. Recently, the SIE/SIES/GITMO consensus criteria for unfitness were validated by the Seattle group in a cohort of 655 adult patients, showing they were able to accurately predict early mortality after intensive chemotherapy, outperforming their TRM model [41]. The Ferrara criteria were also validated by the Network “Rete Ematologica Lombarda” in 699 older AML patients, showing a significant correlation between fitness and survival; median OS for FIT, UNFIT and FRAIL patients was 10.9, 4.2, and 1.8 months, respectively, and the criteria were easily applicable in 98% of patients [11]. Importantly, FIT patients appeared to benefit from intensive chemotherapy, while UNFIT ones showed similar results with non-intensive approaches. Conversely, FRAIL patients did not benefit from any treatment beyond best supportive care. 

**Table 1 cancers-13-05075-t001:** Different models and scores to assess patients’ fitness and eligibility for intensive chemotherapy.

Score	*N*° of Patients	Score Variables	Specificities
	**Early mortality and survival predictive models**		
Malfuson et al. (2008) [26]	416	Age, WBC, PS, cytogenetic risk	Predictive of mortality or survival, not proper fitness scores, therapy-specific, inclusive of disease features
Wheatley et al. (2009) [21]	2483	Age, WBC, PS, cytogenetic risk, type of leukemia (de novo vs. secondary)	
Kantarjian et al. (2010) [23]	446	Age, PS, cytogenetic risk, creatinine	
Krug et al.—AML Score (2010) [24]	1406	Age, body temperature, Hb, platelet count, fibrinogen, type of leukemia (de novo vs. secondary)	
Walter et al.—TRM Score (2011) [22]	3365	Age, PS, platelet count, WBC, peripheral blood blast percentage, albumin, creatinine, type of leukemia (de novo vs. secondary)	
	**Geriatric assessment scores**		
Soubeyran et al.—G8 Score (2008) [30] Soubeyran et al. (2011) [31]	364, 1668	Seven Mini Nutritional Assessment (MNA) items (appetite, weight loss, motricity, BMI, cognition and depression, self-related health, medications), age	Quick and easy to apply, generalizability for cancer patients
Deschler et al. (2013) [33]	195	PS (Karnofsky index), activities of daily living (ADL) and QoL/fatigue	Time consuming
Klepin et al. (2013) [35]	74	Cognition, psychological function, physical function, comorbidity	
Sherman et al. (2013) [34]	101	Comorbidity, physical function, pain	
	**Comorbidity and organ function scores**		
Sorror et al.—HCT-CI Score (2005) [37]	1055 (+ 347)	Comorbidities	Originally developed to assess eligibility for HCT
Sorror et al.—AML Composite Model (2017) [39]	733 (+ 367)	Age, augmented HCT-CI and cytogenetic/molecular risks	Inclusive of disease features, possibly able to identify patients who do not benefit from intensive chemotherapy
Ferrara et al. (2013) [40]; Borlenghi et al. (2021) [11]	699	Age, PS, comorbidities (cardiac, pulmonary, renal, hepatic, infections, mental illness, uncontrolled neoplasia)	Easily and widely applicable, not inclusive of disease features, able to predict benefit from more or less intensive treatments in different fitness groups (fit/unfit/frail patients)

Abbreviations. BMI, body mass index; Hb, hemoglobin; HCT, hematopoietic cell transplantation; PS, performance status; QoL, quality of life; WBC, white blood cells.

## 3. Intensive Approaches for FIT Patients

### 3.1. Induction Chemotherapy

Intensive induction chemotherapy has historically been considered the most effective approach for older fit patients with newly diagnosed AML, although this might not be true in all genetic subtypes of the disease (see below). An intensive induction regimen generally allows the achievement of deeper responses compared to less intensive approaches, which could translate into longer OS and event-free survival (EFS) [42]. With standard “7 + 3” regimens, CR is obtained in about 40% to 60% of older adults, but long-term survival can be achieved by no more than 20% of cases [20]. In recent years, many efforts have been made to improve outcomes in this high-risk population, either (I) administering higher doses of cytarabine (>1000–1500 mg/m^2^) and/or adding purine nucleoside analogues; (II) using new drug formulations; or (III) adding new drugs to standard regimens.

#### 3.1.1. Intensifying Standard Induction Chemotherapy

The role of high-dose cytarabine (HDAC, 2–3 g/m^2^) during induction has been widely investigated in the younger population, with controversial results [43,44]. In older patients, Rolling et al. found that the use of intermediate-dose cytarabine (1 g/m^2^) in induction was associated with a higher CR rate compared to standard doses; however, no survival benefit was observed and the risk of relapse was comparable, possibly because of the difference in consolidation intensity between the two arms [45].

HDAC can be combined with the nucleoside analogue fludarabine with or without idarubicin or G-CSF (FLAG/FLAG-IDA/FLAI regimens), with an increased anti-leukemic activity due to the accumulation of cytotoxic metabolites in the blasts [46]. However, most studies employing these regimens were not randomized and involved younger patients [47]. Cerrano et al. reported a 60% CR rate in 192 older patients (median age 67 years) treated with the FLAI regimen, with 60-day mortality below 10%; a CR rate of 54% and a median OS of 12.6 months were observed in the adverse karyotype group (accounting for one third of patients). Moreover, most patients received a dose reduction of 20% (four-day regimen, FLAI-4, with intermediate dose cytarabine), which remained effective. However, long-term survival remained globally poor even with this approach [48]. These results are in line with data by Kim et al., who found a 56% CR rate in the elderly population treated with an attenuated-dose FLAI regimen, which, however, was associated with a not-negligible 60-day mortality of 21% [49]. Cladribine might also be added to standard induction therapy and the Polish group found that it could be associated with higher CR rates and better OS in a subset of patients with non-adverse karyotype aged 60–65 years [50].

#### 3.1.2. New Drug Formulations 

CPX-351, a combination of cytarabine and daunorubicin in a fixed 5:1 molar ratio encapsulated in a liposome, accumulates and persists in the bone marrow for a longer time than the same drugs administered alone thanks to its formulation, resulting in a higher exposure and uptake in the leukemic blasts. In the phase IIII study that led to the approval of the drug, Lancet at al. showed that CPX-351 was associated with improved composite CR (cCR, in this text it includes CR and CR with incomplete count recovery, with different definitions among trials, including CRi, CRh, CRp) rates (48% vs. 33%) and a significant OS (median 9.56 vs. 5.95 months) benefit compared to standard 7+3 in patients aged 60–75 years with secondary or therapy-related AML [51]. Results after allo-HCT appeared particularly promising, possibly because of the ability of CPX-351 to achieve deeper responses, although further studies evaluating measurable residual disease (MRD) status are required to support this hypothesis. Additionally, while the drug was associated with longer duration of neutropenia, mucosal toxicities were reduced compared with standard chemotherapy [52], with a probable positive impact on post-HCT results. Importantly, these results were recently confirmed in the real-world setting, both by the Italian and the French group, with positive outcomes in patients able to proceed to allo-HCT and a low early-death rate [53,54].

To expand the potential benefit of CPX-351 to a broader patient population, Issa and colleagues tested lower doses of the drug in less-selected populations at high risk for induction mortality. Toxicities were manageable, but cCR rate was lower compared to standard doses, and this approach is not currently recommended [55].

Finally, the category of secondary AML is quite heterogeneous, and the subgroups of patients who could benefit the most from CPX-351 remains to be defined. While a recent post hoc analysis focusing on patients who achieved cCR found that the most notable OS improvements were observed in those with t-AML and those aged >70 years [56], its efficacy in the most difficult category (i.e., *TP53*-mutated or secondary treated AML) remain to be established, with inconsistent results among different reports [53,54,57,58]. Interestingly, secondary-like AML gene mutations [59], which have been established as a negative prognostic factor in patients treated with standard intensive chemotherapy [60], appeared to be associated with favorable outcomes with CPX-351 treatment [54]. 

#### 3.1.3. Incorporating Antibody–Drug Conjugates

Gemtuzumab ozogamicin (GO), an anti-CD33 monoclonal antibody that delivers the cytotoxic drug calicheamicin into leukemic cells [61], has been evaluated in several clinical trials in combination with intensive chemotherapy. Burnett et al. found that adding GO (in a single dose of 3 mg/m^2^) to a 7 + 3-like regimen improved OS and EFS in patients aged >60 [62]. In the Acute Leukemia French Association (ALFA) 0701 trial, enrolling patients aged 50–70 years, Castaigne and colleagues found that the addition of a fractioned dose of GO 3 mg/m^2^ on day 1, 4 and 7 improved OS and EFS, although the OS benefit was not significant [63]. The use of GO was associated with more hemorrhagic events, but with a similar rate of infection and early deaths. An acceptable rate of veno-occlusive disease was reported (4.6% vs. 1.5% in the control arm) [64]. Finally, a meta-analysis by Hills et al. found that the benefit of the addition of GO to standard intensive chemotherapy (either 7 + 3 and FLAG-Ida) was associated with a significant 5-year survival improvement and a reduced risk of relapse in the favorable and intermediate cytogenetic risk categories, but not in the adverse group [65]

#### 3.1.4. Incorporating Targeted Agents

In younger patients (<60 years) with newly diagnosed *FLT3*-mutated AML, the FLT3 inhibitor midostaurin is commonly added to the standard 7 + 3 induction therapy, as this regimen was approved based on the positive results of the RATIFY trial [66], but older patients were excluded from the study.

More recently, Schlenk et al. reported the results of midostaurin added to 7 + 3 induction and consolidation therapy in a trial also including 86 patients aged 61–70 years. The cCR rate was 78%, comparable to that of the younger patients, and OS was significantly prolonged compared to historical controls. The best survival benefit both in younger and older patients was observed in those who underwent allo-HCT followed by midostaurin maintenance, which was not permitted in the RATIFY trial [67]. Despite the positive results, attention should be paid when using midostaurin in the older population, given the relatively high rate of cardiac toxicity (mostly arrhythmia) and pulmonary infection reported in the trial. 

The role of other first-generation FLT3 inhibitors combined with intensive chemotherapy is less established; sorafenib added to 7+3 did not show a survival benefit in elderly patients, both with wild-type and *FLT3*-ITD mutation [68]. In contrast, combinations with selective FLT3 inhibitors, such as quizartinib and gilteritinib, are being actively explored. Pratz et al. found that the addition of gilteritinib to a 7+3 regimen was associated with a cCR rate of 100% and an EFS of 9.7 months in a trial including older patients [69], and a phase III, randomized trial comparing 7+3 combined with midostaurin or gilteritinib in patients with newly diagnosed *FLT3-*mutated AML is now ongoing (NCT04027309).

In *IDH*-mutated AML, the combination of the IDH inhibitors ivosidenib and enasidenib (see below) with intensive chemotherapy (“7 + 3” regimen, followed by high-dose cytarabine) was recently explored in a phase I study [70]. Median age was 62.5 years and 63 in in the ivosidenib and enasidenib cohort, respectively, and cCR rates were 72% and 63%, respectively. Ivosidenib and enasidenib were well tolerated and did not affect time to hematologic recovery after induction, and the incidence of IDH-differentiation syndrome was low (3.3%).

In the broader AML population, the incorporation of the BCL-2 inhibitor venetoclax (see below) has been explored in combination with reduced-dose chemotherapy (i.e., 2+5 regimen) [71], and with fludarabine- or cladribine-based regimes [72,73,74], although in the latter studies only a few older patients were included. The combinations appeared effective with very promising cCR rates and early OS data, but further studies are need given the risk of toxicity in this age group.

### 3.2. Consolidation Chemotherapy

In younger AML patients achieving CR after induction, consolidation therapy (chemotherapy vs. autologous or allogeneic stem cells transplantation) is administered based on the disease risk profile of the patient, the tolerability of further intensive treatment and, more recently, MRD status [20,27,75,76]. Generally, age by itself is not to be considered a limit to the administration of a post-remission treatment. ASH guidelines and NCCN guidelines for the older AML population suggest proceeding with consolidation therapy over no additional therapy in patients achieving CR, following a global evaluation considering performance status, age, associated comorbidity and adverse features, often present in this population [27,42]. However, the small number of prospective studies in older patients makes the optimal strategy unclear. In 1994, Mayer and colleagues demonstrated the unfeasibility of the administration of multiple courses of HDAC in older AML patients [77]. Since then, only a few studies have addressed the question, but globally intensified approaches failed to show a benefit. In a study by Stone et al., the association of cytarabine and mitoxantrone versus cytarabine alone did not improve outcomes, with the former regimen being associated with greater toxicity [78]. Data from the ALFA randomized trials could not show a survival benefit of a more aggressive approach [79,80] and, more recently, the data of 441 patients >60 years in first CR from a single French institution confirmed that the use of standard-dose chemotherapy translated into similar outcomes compared to intensified strategies [81].

### 3.3. Autologous Stem Cell Transplantation

Prospective studies in older AML patients assessing the benefits of autologous SCT compared to chemotherapy consolidation or allogeneic transplantation are lacking. As emerged in previous works, autologous SCT could be safely offered to a selected number of older patients with intermediate–low-risk AML, provided that a consistent number of peripheral blood stem cells (PBSCs) are collected [82,83,84]. As suggested by ELN 2017 guidelines, it could be considered in intermediate-risk patients, and the current GIMEMA strategy is to limit autologous HCT to patients with favorable/intermediate-risk AML and MRD negativity [20,75], although it is mostly reserved for younger patients. 

### 3.4. Maintenance Therapy

Allo-HCT is not a feasible option for a significant proportion of patients above 60 years who obtain a cCR after intensive induction/consolidation chemotherapy [85]. Maintenance therapy, consisting of the continuous administration of low-dose anti-leukemic drugs, could represent an option, since most of these patients are at high relapse risk. However, although with some controversial results [86], maintenance strategies, either low-dose cytarabine (LDAC) or immunotherapy based, have historically failed to show a consistent survival benefit [87]. 

The hypomethylating agent (HMA) azacitidine has been explored in this setting, as the continuous exposure of residual blasts to an epigenetic modulation could reactivate tumor suppressor genes able to re-induce the apoptosis pathway in these aberrant cells. In the HOVON97 trial, 116 older patients with AML/HR-MDS in cCR after intensive chemotherapy were randomized to receive either subcutaneous azacitidine 50 mg/m^2^ for 5 days every 4 weeks subcutaneously or placebo. Disease-free survival (DFS) was significantly improved in the azacitidine arm (64% vs. 42% at 12 months), while OS was not, possibly because of the greater use of rescue therapy in the placebo arm [88]. Conversely, an Italian study evaluating subcutaneous azacitidine maintenance with a 7-day schedule did not show a significant DFS benefit, except possibly in patients aged >73 years, with a higher rate of adverse events in the azacitidine arm [89]. 

Recently, the QUAZAR AML-001 trial evaluated the use of the oral formulation of azacitidine (CC-486, 300 mg daily) as maintenance therapy in 472 older AML patients with intermediate or poor risk cytogenetics (≥55 years, median age 68 years) in cCR after intensive chemotherapy. CC-486 was associated with significantly longer relapse free survival (10.2 months vs. 4.8 months) and OS (24.7 months vs. 14.8 months). Toxicity was acceptable: in the CC-486 arm, SAEs were detected in 33% of patients, with a 17% rate of infection, and 13% of the patients eventually interrupted the experimental drug due to adverse events [90].

To further improve results, a phase III (NCT04102020) randomized trial is evaluating the combination of azacitidine and venetoclax versus best supportive care as maintenance therapy in both younger and older patients in cCR after intensive chemotherapy, but the study is ongoing, and results will not be available soon.

The role of maintenance with targeted agents is not established outside the allo-HCT setting, with a post hoc analysis of the RATIFY finding no difference between patients who received maintenance with midostaurin or placebo in younger patients [91], and a randomized trial testing gilteritinib maintenance vs. placebo being prematurely closed (NCT02927262).

### 3.5. Allogeneic Hematopoietic Cell Transplantation

#### 3.5.1. Indications

The indication to proceed to allo-HCT in first CR is often not straightforward, since both disease and patient characteristics should be taken into account. Indeed, the genetic risk category, commonly according to ELN 2017 risk classification, is essential, but the risk of TRM, which can be very relevant in this age group and that depends on patients’ performance status and comorbidities, should be carefully considered [92]. Additionally, the role of MRD is being increasingly recognized, and it is being integrated into treatment algorithms [76]. In general, allo-HCT in first CR is considered when the relapse risk without such an intensive procedure exceeds 35% and the TRM is acceptable [20,93,94,95].

Allo-HCT is associated with significant morbidity and mortality, particularly in older patients. Since the median age at diagnosis of AML is 67 years, allo-HCT could be precluded to a consistent proportion of patients. However, improvements in supportive care, the introduction of reduced-intensity conditionings (RIC) or nonmyeloablative conditionings (NMA), the increased availability of unrelated donors and the positive results of the post-transplant cyclophosphamide-based platform for haploidentical transplants have broadened the applications of allo-HCT. Indeed, a marked uptick in transplants for patients older than 60 years has been documented in recent years and a 2017 CIBMTR analysis showed that 31% of patients receiving allo-HCT were above 60 years and 6% were older than 70 years [96]. On the other hand, recent data suggest that less than 10% of elderly patients with AML between 60 and 75 years will ultimately undergo transplantation [97], especially if treated outside tertiary care centers [98].

#### 3.5.2. Outcome of Transplantation in Older Patients with AML

Table 2 summarizes the main studies including older adults undergoing allo-HCT for AML. Overall, the survival rate has improved over the years, with estimates ranging from 15% up to 62% at 1–3 years. Nonetheless, transplant-related toxicity remains significantly high. Although the use of RIC and NMA regimens preserve a graft-vs.-leukemia effect, relapse remains the major hindrance to the favorable outcome of older patients undergoing allo-HCT, with estimated rates at 1–2 years of 27–56%. Two meta-analyses investigated the role of HCT in elderly patients with AML/MDS. Zhang et al. analyzed the efficacy and safety of allogeneic HCT with RIC in patients older than 50 years with AML/MDS. Three-year OS, EFS and NRM were 42%, 43% and 27%, and the corresponding figures for patients ≥ 60 years were 42%, 45% and 26%, respectively. Estimates of relapse and grade II–IV acute graft versus host disease (aGVHD) were 33% and 35%, respectively [99]. Rashidi et al. analyzed a total of 13 studies including 749 AML patients >60 years. RFS and OS at 3 years were 35% and 38%, respectively, while 1-year NRM was 26% [100].

#### 3.5.3. Prognostic Models

The HCT-CI, as discussed above, is the most widely used score to predict NRM in patients undergoing allo-HCT, but it poorly correlates with the outcomes of elderly patients [37]. Few models have incorporated age as a potential predictive factor. The EBMT score, including age, disease status, time interval from diagnosis to transplant, donor type and donor recipient sex combination, could improve the selection of HCT candidates [115]. More recently, a Japanese study found that five parameters (namely age, sex, ECOG, HCT-CI and donor type) were associated with NRM and could be used to group patients into low-, intermediate-, high- and very-high-risk categories, with markedly different outcomes after allo-HCT [116].

Generally, age per se should not be considered an exclusion criterion to proceed to allo-HCT. On the contrary, a multidimensional approach should be followed, evaluating a larger spectrum of variables that could impact on HCT outcome. Most of the above-mentioned scores consider comorbidities as a parameter to evaluate the patient’s fitness and performance status (i.e., Karnofsky score and ECOG) as a more reliable indicator of outcome than chronological age [117]. In addition, frailty, which refers to a multidimensional syndrome consisting of a reduction in physiologic reserve that increases vulnerability, could be present in the absence of severe comorbidities in this age group. In a study by Hedge and colleagues, frailty was associated with adverse outcomes after allo-HCT, suggesting that its assessment could be used to implement risk stratification of HCT recipients [118]. Furthermore, CGA can be considered in older patients that are candidates for allo-HCT to identify those at higher risk of complications related to major frailty [119,120]. Indeed, a recent study including 228 patients older than 60 years who underwent allo-HCT evaluated the efficacy of a multidimensional geriatric assessment to stratify them into FIT, UNFIT and FRAIL, and found that this score was highly predictive of survival, outperforming HCT-CI [121].

## 4. Less-intensive Approaches

In patients UNFIT for intensive chemotherapy, lower-intensity therapies used to include LDAC and the HMAs azacitidine and decitabine. Despite the survival benefit compared with supportive care alone, the prognosis of patients treated with HMAs remained unsatisfactory, with cCR rates of 20–30% and median OS usually around 8–10 months [122,123,124]. Recently, a US population-based study including 2263 older HMA-treated AML patients confirmed a median OS of 7-8 months, without a significant difference between azacitidine and decitabine [125]. The advanced understanding of AML mutational landscape clearly outlined the biological differences of the disease in young vs. older patients. In the elderly, AML harbors high-risk mutations more frequently, such as *DNMT3A*, *SRSF2*, *ASXL1*, *RUNX1* and *TP53*, confirming the role of genetics in explaining the poor outcomes in this age group [126,127,128]. On the other hand, the prevalence of *IDH* mutations correlates with increased age, with roughly 20% of patients above 60 years harboring these aberrations [129,130,131]. Thus, targeting IDH would be particularly relevant in older AML patients.

The identification of targetable AML-specific surface markers, driver oncogenes and cellular pathways finally led to the development of new effective treatments; see Figure 1 [132,133] In the last few years, six new drugs have been approved by the U.S. Food and Drug Administration (FDA) for the treatment of newly diagnosed or relapsed or refractory (R/R) AML in UNFIT patients: the FLT3 inhibitor gilteritinib, the IDH inhibitors ivosidenib and enasidenib, the anti-CD33 monoclonal antibody gemtuzumab ozogamicin, the hedgehog pathway inhibitor glasdegib and the BCL-2 inhibitor venetoclax [10]. In this new context, a detailed genetic characterization of the disease with early identification of targetable mutations has become essential. In addition, it is crucial to be able to predict and promptly manage the specific drug complications.

### 4.1. Venetoclax-based Combinations

Early pre-clinical studies have demonstrated that AML cells are highly dependent on BCL-2 for survival, providing rationale for testing the highly selective oral BCL-2 inhibitor venetoclax [134,135,136], which also showed synergistic antileukemic activity with HMAs and chemotherapy in preclinical models [137,138,139].

After the very positive results of a phase Ib clinical trial [140], venetoclax (400 mg daily) was tested in combination with azacitidine vs. azacitidine alone in the international randomized phase 3 VIALE-A trial enrolling 431 older UNFIT AML patients. With a median follow-up of 20 months, azacitidine–venetoclax was associated with a significant OS benefit (14.7 vs. 9.6 months). cCR rate and median duration of response were also significantly improved with the combination (66.4% vs. 28.3% and 13.9 vs. 17.8 months, respectively). Although a benefit was observed in all AML genomic risk groups, it was particularly marked in patients with intermediate cytogenetic risk and/or *IDH* mutations. Conversely, despite a significant improvement in cCR rates, long-term outcomes remained poor in patients with *TP53* mutations and/or adverse cytogenetics. The safety profile of azacitidine–venetoclax was consistent with that reported in previous studies, with a significantly higher frequency of cytopenia and febrile neutropenia compared to treatment with azacitidine alone [141].

Recently, a phase II trial tested the efficacy and safety of 10-day decitabine, an intensified regimen with controversial benefit compared to the standard 5 days [142,143,144], with venetoclax in patients deemed ineligible for intensive chemotherapy with both newly diagnosed and R/R AML. The overall response rate (ORR) was 74% (89% in newly diagnosed AML) and the most common serious adverse events among all patients were neutropenic fever (38%), pneumonia (10%) and sepsis (10%). The median OS in newly diagnosed AML was 18.1 months [145]. 

Following the promising results of a phase Ib/II study [146], venetoclax (600 mg daily) with LDAC was compared to LDAC alone in the phase III randomized VIALE-C trial, which included 211 older AML patients. The addition of venetoclax to LDAC resulted in a 25% survival benefit (median OS 7.2 vs. 4.1), but statistical significance was reached only after an unplanned analysis with an additional 6 months of follow-up. cCR rate was 48% and 13% in the venetoclax + LDAC and LDAC-alone groups, respectively. Of note, patients with previous HMAs were also included, unlike in the VIALE-A trial. This study did not highlight new safety signals, with the main grade 3/4 adverse events in the venetoclax arm being cytopenia and febrile neutropenia (32%) [147].

These results led to an extensive incorporation of venetoclax combination therapies in the clinical setting and to their recognition as a standard-of-care first-line treatment in UNFIT patients with AML, mostly replacing monotherapy with azacitidine or decitabine. 

Recent real-life data have generally confirmed the significantly improved remission rates with HMAs and venetoclax combinations, but non-negligible risks of prolonged hematological toxicities and serious infections exist and should be thoroughly managed [148,149]. In a Mayo Clinic series of 44 newly diagnosed AML patients, with a median age of 73.5 years, the authors reported a cCR rate of 50%, significantly superior to the 23% of a matched AML cohort treated with HMAs alone, but median OS was not improved [150]. A small study by the Mount Sinai group analyzing 26 newly diagnosed AML patients treated with venetoclax in combination with azacitidine or decitabine, between April 2018 and January 2020, reported a cCR rate of 53.8%, but concerns regarding frequent infections, persistent cytopenias and transfusion requirements were raised [151]. 

### 4.2. Glasdegib

Glasdegib is a potent and selective oral inhibitor of the positive Hedgehog pathway regulator Smoothened [152,153], which was tested in a phase I/II trial including previously untreated AML and high-risk MDS patients, in combination with either LDAC, decitabine or intensive chemotherapy, showing CR rates of 8%, 28% and 54%, respectively [154]. A subsequent randomized phase II trial investigated the combination of glasdegib (100 mg daily) with LDAC vs. LDAC alone in 132 patients with AML or high-risk MDS and reported an improved CR rate (17% vs. 2%), and prolonged survival (median OS of 8.8 vs. 4.9 months). The addition of glasdegib to LDAC was generally well tolerated, the most common AEs being cytopenias and gastrointestinal events. Of note, AEs specifically related to glasdegib were muscle spasms, nausea and dysgeusia [155]. Based on this study, glasdegib was approved in combination with LDAC for newly diagnosed AML patients older than 75 years or unfit for intensive chemotherapy, and its combination with HMAs is currently under investigation (NCT03416179).

### 4.3. FLT3 Inhibitors 

Given the lack of effective treatments for *FLT3*-mutated unfit AML patients and the disappointing results of azacitidine alone in this subset of patients [156], FLT3 inhibitors have been tested in combination with HMAs, following the demonstration of their synergistically cytotoxic effect [157]. Sorafenib was tested in combination with azacitidine in a phase II study including 43 patients, mostly pretreated and with a median age of 64 years, showing an ORR rate of 46% and CR rate of 16% [158]. A recent study reported the activity of sorafenib combined with azacitidine in 27 newly diagnosed unfit *FLT3*-mutated AML patients with a median age of 74 years. ORR was 78%, with a CR rate of 26% and cCR rate of 44%. Median OS was 8.3 months and this regimen was generally well tolerated, the most frequent grade 3-4 adverse events being neutropenic fever and infections, both occurring in 26% of patients [159]. In elderly patients, midostaurin has been combined with azacitidine in a phase I/II trial including 54 patients, with median age of 65 years. Only 74% of patients had *FLT3* mutations and only 24% were treatment naïve. The overall response rate was 26% in the entire population and 33% in *FLT3*-mutated patients previously unexposed to other FLT3 inhibitors [160]. Gilteritinib at a dose of 120 mg daily was compared to salvage chemotherapy in a phase III randomized trial including 371 R/R AML *FLT3*-mutated patients, with a median age of 62 years. cCR rate was 34% in the gilteritinib group vs. 15.3% in the chemotherapy one and median OS was significantly longer with gilteritinib (9.3 vs. 5.6 months). Notably, adverse events were less common in the gilteritinib group [161]. Following these results, gilteritinib has been approved in R/R *FLT3*-mutated AML patients. Surprisingly, preliminary data of a phase III trial of gilteritinib combined with azacitidine vs. azacitidine alone (NCT02752035) in newly diagnosed *FLT3*-mutated UNFIT AML patients failed to demonstrate a survival benefit for the addition of the FLT3 inhibitor. However, this trial is not yet published and review of the data is ongoing.

### 4.4. IDH Inhibitors

Ivosidenib was first tested in a phase I dose-escalation and dose-expansion study, enrolling 258 patients with *IDH1*-mutated AML. In the primary efficacy population, which included 125 patients with R/R AML, ORR, cCR and CR rate were 41.6%, 30.4% 21.6%, respectively, with a median duration of response of 9.3 months [162]. The most relevant adverse events were QT interval prolongation, leukocytosis and IDH differentiation syndrome [163]. This phase I study also allowed the enrollment of 34 older patients with newly diagnosed AML ineligible for standard therapy, with a median age of 76.5 years. Among them, the cCR and CR rate were 42.4% and 30.3%, respectively, with a median OS of 12.6 months after a median follow up of 23.5 months. IDH differentiation syndrome was reported in 18% of patients (grade 3 or higher in 9%) but did not require treatment discontinuation [164].

Enasidenib was tested in a pivotal phase I/II study enrolling 239 patients with mutant *IDH2* and advanced myeloid malignancies [165]. In R/R AML patients, CR rate was 19.6% and ORR was 38.8%, with a median OS of 8.8 months. The most relevant grade 3-4 treatment-related adverse events were hyperbilirubinemia and IDH differentiation syndrome. Thirty-nine older patients with newly diagnosed *IDH2*-mutated AML not candidates for intensive therapy were also enrolled, with a median age of 77 years. ORR was 30.8% and CR rate was 18%. After a median follow-up of 8.4 months, median OS was 11.3 months [166]. Following these results, ivosidenib (500 mg daily) and enasidenib (100 mg daily) both received FDA approval for adults with R/R *IDH1*- or *IDH2-*mutated AML, respectively, while only ivosidenib has been approved in newly diagnosed UNFIT patients.

The combination of IDH inhibitors with HMAs showed synergistic activity, and clinical data in patients unfit for intensive chemotherapy are promising [167,168,169]. A phase Ib trial tested the combination of ivosidenib plus azacitidine in 23 newly diagnosed *IDH1*-mutated AML patients ineligible for intensive chemotherapy, showing an ORR and CR rate of 78.3% and 60.9%, respectively, and a 12-month survival of 82% [170]. The most common adverse events were QT prolongation (26%) and IDH differentiation syndrome (17%), consistent with the safety profile reported for ivosidenib in monotherapy. Importantly, severe myelotoxicity was not frequent, with treatment-related grade ≥3 neutropenia, anemia and thrombocytopenia occurring in 22%, 13% and 13% of patients, respectively. Enasidenib has been combined with azacitidine in a randomized phase I/II study in newly diagnosed *IDH2*-mutated AML patients ineligible for intensive chemotherapy. Preliminary data demonstrated improved CR rates (53% vs. 12%) in the combination arm compared to azacitidine alone, while a non-significant trend towards a longer EFS was observed (17.2 vs. 10.8 months) without survival benefit. Although neutropenia was more common in the combination arm (35% vs. 22%), grade 3–4 infections were more frequent in the azacitidine alone arm (31% vs. 18%, respectively). IDH differentiation syndrome occurred in 12 patients (18%) treated with enasidenib [171].

### 4.5. Future Perspectives

#### 4.5.1. APR-246

The outcome of *TP53*-mutated AML has consistently been unfavorable, with poor survival regardless of the treatment choice [172]. Most *TP53* missense mutations result in a misfolded transcription factor not able to bind DNA, allowing tumor cells to evade apoptosis and proliferate. Eprenetapopt (APR-246) is a p53-reactivating agent that restores the normal folding of mutant p53 by binding key cysteine residues and is the first mutant p53-targeting compound in clinical development (see Table 3) [173]. The results of a phase Ib/II study combining APR-246 with azacitidine in *TP53*-mutated MDS/AML have been presented. Among the 55 patients enrolled, ORR was 71% and CR rate 44%. Patients with isolated *TP53* mutations showed higher rates of CR (69% vs. 25%) and responding patients had significant reductions in *TP53* VAF, with 38% achieving complete molecular clearance. With a median follow up of 10.5 months, median OS was 10.8 months, similar between MDS and AML patients. The most common grade ≥ 3 adverse events were febrile neutropenia (33%), leucopenia (29%) and neutropenia (29%) [174]. A phase II study by the Groupe Francophone des Myélodysplasies (GFM) tested eprenetapopt in combination with azacitidine in untreated high- or very-high-risk *TP53*-mutated MDS (34) and AML (18) patients. In MDS patients, an encouraging ORR of 62%, with a CR rate of 47%, was observed, while in AML, ORR and CR rate were 33% and 17%, respectively, with no patient with more than 30% bone marrow blasts obtaining CR. The main toxicities were febrile neutropenia (36%) and neurologic adverse events (40%), the latter correlating with a lower glomerular filtration rate at treatment onset and higher age [175]. 

#### 4.5.2. Pevonedistat

Pevonedistat is a first-in-class NEDD8-activating enzyme (NAE) inhibitor, implicated in the ubiquitin ligase-proteasome-mediated degradation of several substrates, which results in the disruption of cell cycle progression and cell survival. A phase 1b study combining azacitidine and pevonedistat in 64 elderly, newly diagnosed AML patients showed a CR rate of 39% and a median duration of response of 8.3 months. The treatment was well tolerated, with transient elevation in liver enzymes being the dose-limiting toxicity [176]. Recently, the results of a randomized phase II trial testing pevonedistat plus azacitidine versus azacitidine alone in 120 patients with higher-risk MDS or low-blast AML has been published. Pevonedistat + azacitidine demonstrated a not significant trend towards an increased OS (21.8 vs. 19.0 months) and EFS (21.0 vs. 16.6 months) compared with azacitidine alone. In the 34 low-blast AML patients, median OS trended longer with pevonedistat + azacitidine vs. azacitidine (23.6 vs. 16.0 months), although no ORR improvement was observed [177]. The data of the phase III trial testing this combination in low-blast AML and MDS (NCT03268954) are awaited and a randomized phase II trial exploring the “triplet” of azacitidine, venetoclax and pevonedistat vs. azacitidine and venetoclax in newly diagnosed unfit AML patients (NCT04266795) is currently ongoing. 

#### 4.5.3. New HMAs

Recently, new HMAs with prolonged half-life have been proposed in order to improve outcomes. Guadecitabine is a dinucleotide of decitabine and deoxyguanosine, administered sub-cutaneously and developed to resist cytidine deaminase degradation, which has shown activity in both naive and R/R AML patients, with CR rate of 50% and 23%, respectively [178,179]. However, the randomized phase III study comparing guadecitabine to the physician choice in newly diagnosed AML unfit patients (ASTRAL-1) failed to demonstrate any improvement in remission rate or survival. 

ASTX727, an oral combination of decitabine with the CDA inhibitor cedazuridine, has recently been FDA approved for MDS [180] and is currently being tested in a phase III trial compared to decitabine in MDS and low-blast-count AML (NCT03306264). Moreover, the safety and efficacy of ASTX727 in combination with venetoclax are being investigated in AML (NCT04657081).

#### 4.5.4. New Families of Epigenetic Drugs

Epigenetic alterations are extremely frequent in AML, as a result of somatic mutations in regulators of DNA methylation (*DNMT3A*, *IDH1/2*, *TET2*) and histone acetylation (*ASXL1*, *EZH2*), along with translocations involving epigenetic factors (*KMT2A* fusions). Consequently, epigenetic drugs might be effective in AML and particularly useful in older patients, considering the higher prevalence of epigenetic aberrations with advancing age [181].

Pracinostat is a potent oral pan-histone deacetylase inhibitor, which acts by restoring the expression of tumor suppressor genes. It was studied in combination with azacitidine in a phase II trial including 50 older unfit AML patients, showing an encouraging CR rate of 52% and a median OS of 19.1 months [182]. However, the confirmatory phase III randomized trial (NCT03151408) was terminated prematurely due to lack of efficacy. 

BET (bromodomain and extraterminal) proteins are chromatin readers that allow the transcription of many oncogenes, such as *KMT2A* (or *MLL*). In a phase I study in R/R AML, the BET inhibitor OTX015 showed a modest but clinically significant activity in some patients [183]. Notably, preclinical data suggest an effective role of these agents in specific subtypes of AML (*KMT2A* fusions, *NPM1* or chromatin/spliceosome genes mutations), while others, such as *TP53* AMLs, seem to be resistant [184].

Pinometostat is a first-in-class inhibitor of the DOT1L (disrupter of telomeric silencing 1-like) histone methyltransferase, an enzyme involved in the survival and proliferation of mixed lineage leukemia (*MLL*)-rearranged leukemia cells. It was recently tested in a phase I study including adult patients with advanced acute leukemia, predominantly with 11q23 translocations. It showed some clinical activity as single agent, including two CRs in t (11;19) cases, supporting the therapeutic potential for targeting DOT1L in *MLL*-r leukemia [185]. More potent and selective DOT1L inhibitors are currently tested in preclinical studies.

Iadademstat (ORY-1001) is a small oral compound which acts as a highly selective inhibitor of the epigenetic enzyme lysine specific demethylase 1 (LSD1). LSD1, through its demethylase activity on lysine residues of histone tails, contributes to the differentiation arrest in some molecular subgroups of AML, such as *KMT2A* fusions. In a recent phase I study in R/R AML, iadademstat was well tolerated and demonstrated clinical and biologic activity [186], encouraging the development of a phase II trial designed to test this agent in combination with azacitidine (EudraCT 2018-000482-36).

#### 4.5.5. HMA and Venetoclax Backbone 

The excellent results of HMA and venetoclax led to the evaluation of this combination as a backbone for associations with other drugs, which are under investigation in ongoing clinical trials [187]. Currently approved drugs, such as IDH and FLT3 inhibitors, and the CD33 monoclonal antibody–drug conjugate GO, are being tested, as well as immune checkpoint inhibitors and newer agents targeting different cellular pathways. If the results of these studies will show a further improvement in remission rates and survival, they could open an era of “triplets” for the treatment of AML patients not eligible for intensive chemotherapy.

#### 4.5.6. JAK2 Inhibitors

JAK2 inhibitors are currently approved for intermediate or high-risk myelofibrosis and for polycythemia vera resistant to hydroxyurea. Recently, these drugs have been explored for the treatment of post-MPN AML, an aggressive form of secondary leukemia with poor response to current treatments and dismal prognosis [188].

Considering the frequent involvement of epigenetic modifiers in post-MPN AML, combination strategies of JAK2 inhibitors and classical or new epigenetic agents have been evaluated. A prospective phase I trial tested the association of 5-day decitabine with escalating doses (10 mg, 15 mg, 25 mg, 50 mg) of ruxolitinib for patients with post-MPN AML. A response rate of 57% (with a cCR rate of 23.5%) and a median OS of 7.9 months were observed, and the combination was well tolerated. However, an increased hematologic toxicity was associated with higher doses of ruxolitinib [189].

Based on these findings, a multicenter phase 2 study evaluated decitabine in combination with ruxolitinib at the dose of 25 mg bid for induction and 10 mg bid for subsequent cycles, showing an ORR of 44% with a median OS of 9.5 months [190]. In vitro studies showed that the combination of JAK2 and BET inhibition resulted in enhanced cell apoptosis and the growth arrest of blastic cells from post-MPN AML [191], while in preclinical animal models of myelofibrosis it was able to reduce fibrosis and prolong survival [192]. These findings provided the rational for further testing this combination in high-risk MPN in currently ongoing trials.

Given the high frequency of *IDH1/2* mutations in AML secondary to MPN, IDH inhibitors could play a significant role in this setting. Moreover, in preclinical models of *JAK2/IDH*-mutant MPNs, the combination of JAK2 and IDH inhibitors showed a strong inhibition of tumor growth and suppression of abnormal 2-hydroxyglutarate production [193]. Recently, a retrospective analysis of IDH1/2 inhibitors used in combination strategies (including with ruxolitinib) showed promising results, with complete remission achieved in 3/7 patients who also reached a median OS of 19 months [194].

#### 4.5.7. Immunotherapy

Older patients unfit for intensive chemotherapy are generally not eligible for allo-HCT, the most effective form of immunotherapy in AML, but a wide range of different approaches exploiting the immune system are being investigated. Currently, the only approved drug is the anti-CD33 antibody–drug conjugate GO. Even though in Europe, GO is approved only for fit patients in combination with intensive chemotherapy, it has also been tested as single agent in an older population [195]. Indeed, two monotherapy regimens (i.e., 6 mg/m^2^ on day 1 and 3 mg/m^2^ on day 8 of induction and 2 mg/m^2^ on day 1 every 4 weeks in newly diagnosed patients and 3 mg/m^2^ on days 1, 4 and 7 in relapsed/refractory AML) were approved by the FDA.

Immune checkpoint inhibitors have not shown sufficient efficacy as single agents in AML. However, several clinical trials exploring their association with HMAs have been started, given the increased expression on the AML blasts of CTLA4, PD-1 and PD-L1 after the administration of HMAs [196]. A single-arm phase II trial, including 70 R/R AML patients, investigated the combination of the anti-PD-1 nivolumab with azacitidine. The ORR was 33%, with a cCR rate of 22%. Median OS was 6.3 months, with grade 3 or 4 immune-related adverse events observed in 11% of the patients [197]. Daver et al. explored the synergistic effect of ipilimumab, nivolumab and azacitidine in R/R AML. Preliminary data of this “triplet” therapy suggested that it is feasible, but it was associated only with a modest OS improvement compared to historical HMA controls [197].

Magrolimab is a monoclonal antibody anti-CD47, a macrophage immune checkpoint, which showed activity against leukemia stem cells inducing tumor phagocytosis. Magrolimab was tested in association with azacitidine in a phase 1b study in AML patients unfit for intensive chemotherapy, showing a promising CR rate above 40%, also in *TP53*-mutated AML, and no immune-related adverse events were observed [198].

Cusatuzumab is a monoclonal antibody directed against CD70, the tumor necrosis factor alpha receptor ligand, which is expressed at the surface of AML stem cells. Since HMAs were shown to increase CD70 expression on leukemic stem cells, cusatuzumab was explored in combination with azacytidine in a phase I/II trial, showing a cCR rate above 80% with no dose-limiting toxicities reported [199].

A different immunotherapy approach involves bispecific antibodies binding both tumor and effector T lymphocyte cells to induce MHC-independent killing. Flotetuzumab is a bispecific antibody-based CD3-CD123 molecule belonging to the dual-affinity retargeting antibody (DART) class. It has been tested in R/R AML patients in a phase I/II study, showing a relatively modest CR rate of 18%. Notably, a 10-gene signature was able to accurately predict complete responses to flotetuzumab [200]. Moreover, *TP53*-mutated AML showed remarkably high response rates with this drug, probably due to the enhanced immune infiltration [201]. Consistent with the known toxicities of bispecific antibodies, the most common adverse events related to flotetuzumab were infusion reactions and cytokine release syndrome, in some cases requiring treatment with steroids and tocilizumab [200]. Several other immunotherapy strategies are currently under investigation, including vaccines and CAR-T cells [202], and they might represent innovative future options for the treatment of older AML patients. 

**Table 3 cancers-13-05075-t003:** Selected new drugs currently being tested in AML.

Agent	Target	Study Phase	Association	AML Population	Age in Study, Median (Range)	cCR in AML	References
Eprenetapopt	p53	Ib/II	Azacitidine	*TP53*-mutated MDS/AML	66 (34–85)	54%	[174]
		II	Azacitidine	Untreated *TP53*-mutated MDS/AML	74 (44–87)	17%	[175]
Pevonedistat	NEDD8 activating enzyme	II	Azacitidine	Higher risk MDS/low-blast AML	72 (34–91)	41%	[177]
		II	Azacitidine + venetoclax	Newly diagnosed unfit AML	NA	NA	NCT04266795
		III	Azacitidine	Higher risk MDS/low-blast AML	NA	NA	NCT03268954
Guadecitabine	Aberrant DNA methylation	II	Alone	Newly diagnosed unfit AML	77 (62–92)	54%	[178]
		I	Alone	R/R MDS/AML	68 (36–86)	6%	[179]
		III	Alone	Untreated unfit AML	76	19.4%	NCT02348489
ASTX727	Aberrant DNA methylation	III	Alone	MDS/ low blast AML	NA	NA	NCT03306264
		I	Venetoclax	Newly diagnosed unfit AML	NA	NA	NCT04657081
Pracinostat	Histone deacetylase	II	Azacitidine	Newly diagnosed unfit AML	75 (66–84)	52%	[182]
		III	Azacitidine	Newly diagnosed unfit AML	NA *	NA *	NCT03151408
OTX015	BET	I	Alone	R/R AML	70 (60–75)	7%	[183]
Pinometostat	DOT1L	I	Alone	Advanced AML	50 (19–81)	4%	[185]
Iadademstat	LSD1	I	Alone	R/R AML	67 (30–81)	3.7%	[186]
		II	Azacitidine	Newly diagnosed unfit AML	NA	NA	EudraCT 2018-000482-36

* study stopped prematurely due to a lack of efficacy. Abbreviations. cCR, composite CR; R/R AML, relapsed/refractory AML; MDS, myelodysplastic syndrome.

## 5. Precision Medicine

The “one-size-fits-all” paradigm has definitely come to an end in AML and, beyond the evaluation of the fitness required to receive intensive treatment, it is now of paramount importance to obtain a detailed genomic characterization in each patient. Indeed, time from diagnosis to treatment start did not impact on patient outcome [203,204], allowing a wait for the results of biological testing, including for most patients with hyperleukocytosis [205].

Not only can a detailed genetic characterization allow the use of a drug specifically targeting a mutated pathway, but it is essential to define the most appropriate therapeutic strategy. Indeed, cytogenetics has long been recognized as the main determinant of patient outcomes after intensive chemotherapy. For instance, patients with core binding factor AML, a rare entity in the elderly, have a significant chance of achieving a prolonged survival with intensive chemotherapy, although results are globally inferior to younger patients [206,207]. Conversely, patients with complex karyotype, especially if also monosomal [208], have shown uniformly very poor outcomes, questioning the role of intensive chemotherapy even in cases deemed fit to receive it. Additionally, the prognostic and predictive role of molecular abnormalities has clearly emerged (e.g., relatively good chemosensitivity of *NPM1-*mutated AML, dismal results in *TP53*-mutated ones, recently reviewed in [172]), but their interaction with cytogenetics is complex, and should also be carefully considered. To address this issue more systematically, the ALFA group analyzed the outcome of 471 patients older than 60 years with a detailed molecular and cytogenetic characterization treated in the ALFA1200 study, with 7 + 3 induction and intermediate-dose cytarabine consolidation. Combining gene mutations and cytogenetic aberrations, the authors were able to identify a subgroup of patients who achieved long-term survival with intensive chemotherapy, namely those with non-poor cytogenetics, *NPM1* mutation and at most one mutation among *FLT3*-ITD low allelic ratio, *DNMT3A*, *ASXL1* or *NRAS* or non-poor cytogenetics and *NPM1*, *FLT3*-ITD, *DNMT3A*, *ASXL1* and *NRAS* being all wild type. Conversely, those with poor-risk cytogenetic with *KRAS* and/or *TP53* mutations showed dismal outcomes (2 years OS 2.8%), suggesting other approaches are strongly needed in this subset [209].

Another strategy that could be helpful to balance the impact of disease genetics and of patient characteristics, to optimize personally tailored therapeutic strategies, is the “knowledge bank approach” proposed by Gerstung and colleagues, which also takes into account clinical parameters such as leukocytosis [210]. Using matched genomic–clinical data from over 1500 AML patients [211], the authors were able to estimate the survival and the probability of different causes of mortality in each case (i.e., TRM, death after relapse, death without relapse), and to evaluate the role of allo-HCT. This “knowledge bank” approach was validated in a real-life cohort [212], and was recently integrated by the French group with the ELN 2017 risk stratification and MRD response to optimize indications of allo-HCT in CR1 [76]. However, knowledge banks require frequent updating because, as new effective treatments become available, survival estimates could become inaccurate if based on the data of patients treated with traditional chemotherapy programs, and all the established prognostic factors should ideally be considered, including those which were recently discovered [213,214,215].

The clinical applicability of a genomic-based approach to treat AML has been recently tested in the US Beat AML trial. A total of 487 older AML patients have been enrolled so far, with comprehensive genetic data available within 7 days before treatment assignment. Patients were enrolled in a sub-study based on their genetic profile and fitness. Eventually, more than half of the patients could be enrolled in a Beat AML sub-study, showing a significant OS benefit compared to those receiving standard treatment. Despite not being randomized, this study confirmed for the first time the validity of a precision medicine approach in these patients [216].

Finally, biological data from the Beat AML trial showed that response to drugs, assessed by ex vivo drug sensitivity assays, could be predicted by the genetic landscape of the disease and by gene expression profile [217], and different drug screening platforms are being implemented by several groups to find the best drug combinations given a specific patient’s profile [218]. Although these tools are not routinely available yet, they will likely help to optimize the treatment possibilities for older AML patients.

## 6. Quality of Life and Palliative Care

The diagnosis of AML is a dramatic event associated with a high physical and psychological burden. In older patients, age-related frailties and social conditions pose further challenges concerning QoL and health care. Published data showed that older patients with AML spend 28% of their time in the hospital and 14% attending outpatient clinic appointments [219]; in addition, almost one-third are admitted to the intensive care unit at diagnosis or after subsequent treatments [220]. However, even if patients with aggressive hematologic malignancies have high symptom burden and impaired QoL comparable to those of solid tumor patients [221,222], the use of palliative care services is less frequent in hematologic settings [223]. Moreover, end-of-life quality outcomes are worse in hematologic patients compared to those with solid tumors [224,225]. Indeed, hematologic malignancies have unique characteristics that make it difficult for clinicians to promptly identify refractory disease or end-of-life stages and this may lead to a delay in the use of palliative care [226,227]. Therefore, QoL assessment and palliative care interventions represent an area of unmet needs and required improvements. To reach this goal, clinicians should address patients’ needs at diagnosis and during disease course, and better integrate palliative care in the health care process. 

### 6.1. Older AML Patients’ Needs

When asked, the majority of older patients report to value a greater QoL over length of life [228,229], suggesting that it is crucial to consider both aspects, taking into account individual feelings and preferences. Addressing AML patients’ needs, Boucher et al. showed that they feel a remarkable psychological distress around time of diagnosis, with troubles in disease understanding and acceptance [230]. Later, during the course of the disease, patients reported fatigue with restrictions in daily activities more frequently than pain, and psychological issues, such as feelings of “helplessness” or “hope-lessness”, depression and uncertainty regarding prognosis and the next steps in treatment [230,231,232]. In addition, patients spoke about experiencing isolation and loss of independence as a consequence of feeling ill, and caregiver burden was a concern for many of them [230,233]. Finally, prognostic misconception due to discordance in patients’ and physicians’ expectations is a common cause of distress for patients [234].

### 6.2. Health-Related Quality of Life Assessment

Health-related QoL (HRQoL) is a multidimensional concept that includes a variety of domains related to physical, mental, emotional and social functioning, each of which may be affected by the disease. Several tools are available to assess QoL. Approximately, they can be divided into general, cancer-specific and leukemia-specific questionnaires. Unlike many advanced solid tumors, where declines in daily functioning occur relatively slowly, AML usually has rapid onset and, therefore, the use of leukemia-specific questionnaires may help to better capture the particular experience of AML patients [228]. Available tools include the EORTC QLQ Leukemia Module (EORTC QLQ-Leu) [235] and the FACT Leukemia (FACT-Leu) [236]. In addition, there are several symptom-specific questionnaires such as the MD Anderson Symptom Inventory for AML (MDASI-AML) providing a comprehensive assessment of symptoms related to AML [237]. Recently, the role of patient-reported outcomes (PROs) questionnaires in predicting survival in AML older patients was addressed by Peipert et al. The authors, using data from one of the largest trials reporting HRQoL in older AML patients according to FACT-leu scale (AML2002), found that PROs were independent predictors of OS among AML patients unfit for intensive therapy. These findings confirmed data already emerged in patients with solid tumors and underscore the need to more systematically collect PRO data in routine care in the elderly AML population [238].

Furthermore, HRQoL models based on condition-specific preference-based measures (CSPBMs) might be able to generate health utilities for economic research [239]. Recently, a preference-based scoring model, the EORTC Quality of Life Utility Core 10 Dimensions (QLU-C10D), has been proposed as a cancer-specific PBM useful as a research tool in health economic evaluation. With 10 HRQoL domains, and four severity levels per domain, it is particularly suitable to detect health differences and generate health utilities for economic research in myelodysplastic syndromes. This model might also provide useful insights for AML in the elderly, which frequently evolves from a previous MDS condition [240].

### 6.3. Integrated Palliative and Oncology Care 

The term “palliative care” was historically used synonymously with end-of-life care. Instead, in a modern conception, palliative care is considered an approach that improves the QoL of patients and their families facing a life-threating illness and should be provided independently of prognosis, even concurrently with active cancer treatment [241]. In the oncologic/hematologic setting, the concept of “palliative care” is often a source of confusion. To clarify the issue, it is essential to differentiate between “primary palliative care”, which alludes to the many interventions usually provided by oncologists/hematologists in clinical practice (initial care of pain and non-pain symptoms, basic management of depression and anxiety, discussions of prognosis and goals of treatment) and “specialty palliative care”, which refers to more advanced interventions performed by board-certified palliative care specialists (management of refractory pain, management of complex mood and anxiety disorders and existential distress, conflict resolution between families and medical staff) [242]. In a study conducted by El-Jawahri et al., early integration of specialty palliative care and oncology care was shown to improve QoL and psychological burden in AML patients treated with intensive chemotherapy, while the authors were not able to demonstrate a significant difference in symptom burden [243]. The evidence of this trial, together with previous data from hematologic patients undergoing stem cell transplantation [244,245], support early specialty palliative care integration into routine clinical care for patients with high-risk hematologic cancers, particularly those enduring prolonged hospitalization. 

### 6.4. Supportive Care 

Supportive care, inclusive of blood product transfusions, anti-infective prophylaxis and treatment toxicities management, is equally essential in AML management. In fit patients receiving intensive chemotherapy in a hospital setting, antifungal prophylaxis is recommended during aplasia, while antibiotic prophylaxis is not uniformly employed because of the worldwide increasing problem of antibiotic resistance [246,247,248]. Since older patients may experience prolonged hematologic recovery, growth factors could be considered as part of supportive care, especially if patients experience serious infections. Transfusion with red blood cells is generally recommended for a hemoglobin level ≤7–8 g/dL, but a higher threshold (> 9 g/dL) should be considered in older patients with heart disease or if anemia symptoms are present. Prophylactic platelet transfusions are recommended when platelet count drops below 10,000/µL [20,247] while if fever or other bleeding factors are present platelets should be replaced if <20,000/µL.

Unfit patients receiving less intensive therapies are usually treated as outpatients, although hospitalization can be considered for those receiving HMA + venetoclax during the first cycle. Planning a shared therapeutic approach with patients and caregivers is crucial in this setting. Moreover, avoiding hospitalization is valued as an important aim of the care project. Common treatment toxicities such as nausea, constipation or diarrhea should be prevented or rapidly treated. Anti-infective prophylaxis should be individualized depending on neutrophil count, anti-leukemic drug administered and patients’ comorbidities [249]. 

### 6.5. End-of-Life Care

The dying trajectory of patients with hematologic malignancies is typically characterized by the alternation between acute exacerbations and phases of stable disease in which highly specialized therapies can still be administered, eventually with a rapid dying phase once in the terminal period of the disease [250]. More than half of older AML patients spend the whole period of their final month of life in hospital (51%) and die in hematology units (53%), with another 7% of deaths occurring in intensive care units; only 30% of patients die in palliative care settings [251]. The intensive level of medical care required for dying AML patients, which includes blood product support, management of infective and bleeding complications, administration of antimicrobial agents and central line care, makes it more challenging to optimally provide home–hospice palliative care [252]. However, improvements in this field should be pursued and a rise in the number of patients receiving end-of-life care at home with the involvement of a palliative care specialist is desirable. 

The main cause of death in older AML patients is infective complications [251]. However, the use of antibiotics in the palliative care setting is controversial. Indeed, although some studies demonstrated that antibiotics did not improve the survival of patients under palliative intent of treatment [253], it has never been proved whether these results could be generalized to patients with hematological cancers who are at higher risk of serious infections. Bleeding complications are another frequent event in older AML patients during the last phase of the disease. Hemorrhagic episodes are responsible for about 14% of the deaths in this setting [251]. In order to avoid hemorrhagic complications, regular platelet transfusion is common, with about 50% of patients receiving platelet transfusion in their last week of life [252]. However, prophylactic platelet transfusions show limited benefit in the palliative context and should be administered only in case of clinical signs of bleeding. Instead, red blood cell transfusions may improve fatigue and QoL and therefore should be available in an end-of-life care setting [42].

## 7. Conclusions

The profound knowledge of AML biology, the availability of new effective treatments (see Table 4) and the increased awareness of the heterogeneity of the elderly in terms of fitness and personal expectations have made the current treatment paradigm of older AML patients very challenging. 

First, it is extremely complex to predict the benefit to patients of receiving intensive chemotherapy. Some models are more suited to define patients’ fitness before treatment, while others have been developed to predict survival after intensive chemotherapy to anticipate outcomes, and none are universally accepted. As all the proposed fitness scores are imperfect, information derived from different models could be combined in an individualized approach for each patient, and the clinical experience of the treating physician remains essential [9]. Furthermore, several potentially fit patients would not derive a benefit from intensive approaches compared to less intensive ones because of the genetics of their disease. Indeed, a sub-analysis of the AZA-AML-001 and some, but not all [254], subsequent real-life studies have already shown that patients above 65 years deemed fit to receive intensive chemotherapy globally obtained comparable results with azacitidine vs. chemotherapy [124,125,255,256], although clearly some subgroups can benefit from a certain approach over another [257]. Finally, as more and more “less intensive” approaches are becoming available, with different toxicity profiles and biological activity, even for unfit patients treatment choices are becoming extremely difficult.

Figure 2 summarizes the treatment approach we follow in 2021 outside clinical trials, which will probably be updated often as new therapeutic options and prognostic tools rapidly become available. Importantly, the age cut-offs remain somehow arbitrary, and the number of factors involved in this complex therapeutic decision cannot be exhaustively depicted in any algorithm, which nonetheless could be helpful to define the strategies generally followed in clinical practice.

Patients below 65 years are very often treated as younger ones and can be frequently offered the same clinical trials, usually with dose reductions (e.g., cytarabine dose < 1–1.5 g/m^2^). Based on the disease characteristics, these patients can be offered 7+3 and GO, 7+3 and midostaurin or CPX-351. Patients who are not eligible for those treatments can receive 7+3 or FLAG-ida regimens.

Patients above 65 years up to the age of 75 can be considered for intensive chemotherapy based on their fitness, which we commonly assess using the Ferrara criteria [11] and the G-8 score [18]. In patients deemed FIT, we favor intensive chemotherapy in case of favorable (cyto)genetics, although we recognize that favorable long-term outcomes can be achieved with HMA+ Venetoclax, for instance in *NPM1*-mutated cases. Conversely, we consider intensive chemotherapy in patients with unfavorable genetics only if potentially eligible for allo-HCT, as results with chemotherapy only are usually dismal [258]. Thus, a thorough discussion with the transplantation team is required, since a careful selection is essential to prevent excessive risks of toxicity. In potential candidates we chose either CPX-351, if eligible, or FLAG-Ida, the latter often with dose reductions. In patients particularly unlikely to obtain a remission with any chemotherapeutic approach, such as those with a monosomal karyotype and/or *TP53-*mutations, a treatment with HMA + ventoclax as a bridge to transplant could be considered, although further data to validate this strategy are needed [259]. Unfavorable-risk patients a priori excluded from allo-HCT are offered HMA + venetoclax even if deemed fit for intensive chemotherapy. Intermediate-risk cases are often treated with intensive approaches, followed by allo-HCT, if feasible. Alternatively, after 2-3 cytarabine-based consolidation cycles, maintenance with HMA can be appropriate. When available, oral azacitidine will certainly be offered, possibly also in the favorable-risk group, as *NPM1*-mutated patients seem to derive a particular benefit [260]. In this age group, most patients not eligible for intensive chemotherapy will be offered HMA + venetoclax as the standard of care. We usually prefer this approach to LDAC + venetoclax, as data supporting the former combination are more robust.

Although several groups have used intensive chemotherapy far beyond 75 years of age before the advent of new drugs [126], in the present era we hardly ever consider it, given the high induction death risk, the limited long-term survival and the effective alternative options available. A possible exception might be the rare cases with core binding factor AML if in very good clinical conditions. While many patients above 75 years could still receive HMA + venetoclax, this combination is certainly not appropriate in every case, as this treatment is associated with significant toxicities, certainly superior to HMA alone. Therefore, in unfit patients at high risk of (infective) complications, we think the use of HMA alone may still be a valid option. Although this approach is not recommended by present guidelines and is possibly less effective [148], selected patients could be offered HMA alone first because of toxicity concerns, and, if the treatment is well tolerated, venetoclax can be added after two or three cycles. Disease presentation and its genetics can be important in this context as well, in order to carefully weigh the risks and benefits of each option. In a patient we consider at significant risk of toxicities, we feel more prone to use HMA alone if the disease presents with characteristics predicting that a meaningful benefit of the treatment could be obtained, such low blast count and non-proliferative features [257,261], or if we expect a very short survival despite the addition of venetoclax, such as *TP53*-mutated ones [141]. Conversely, in AML subtypes clearly benefitting from the combination treatment compared to HMA alone, such as *NPM1*- or *IDH*-mutated cases [141,149,262], we would generally favor the combinations, despite the risks. Finally, questions have been raised concerning the cost effectiveness of azacitidine and venetoclax when considering the whole population of unfit AML patients, a further reason to carefully select patients who can benefit the most from the combination of treatment, especially in public health systems [263].

Table 5 summarizes the clinical and biological factors we consider when choosing between intensive chemotherapy, HMA + venetoclax or HMA alone, none of which would ever be sufficient per se, but could be helpful in making the most appropriate decision. Ivosidenib or enasidenib with or without azacitidine or the combination of FLT3 inhibitor sorafenib and HMA could be considered in selected cases not deemed fit to received azacitidine and venetoclax and presenting *IDH1*, *IDH2* or *FLT3*-ITD mutations, respectively. However, a randomized comparison with HMA and venetoclax is not available, and their use remains off label, except for single-agent ivosidenib in the US. The combination of LDAC + glasdegib could also be considered in patients deemed too frail to receive HMA + venetoclax, but data are scarce and we have limited clinical experience so far. Certainly, the logistic advantage of LDAC compared to HMA can favor this option in some contexts.

Importantly, the available treatment options should be thoroughly discussed with patients, as shared decision making in this age group is essential, considering hardly any randomized controlled trial firmly establishing the benefit of a strategy over another is available. Some older patients might refuse the prolonged and difficult hospitalization required to receive intensive chemotherapy [264], while others could prefer a short-term treatment program compared to a life-long one. Finally, several patients, especially above the age of 85 years, are very frail, thus still being candidates for best supportive care only, a context in which expert palliative care is essential.

In conclusion, the revolution of the therapeutic scenario has made treatment decisions extremely complex in older AML patients. As the field is evolving with extreme rapidity, therapeutic algorithms and our way of thinking should be prone to rapid changes to offer the best options available to our patients, since long-term outcomes, despite the enormous advances, remain unsatisfactory.

## Figures and Tables

**Figure 1 cancers-13-05075-f001:**
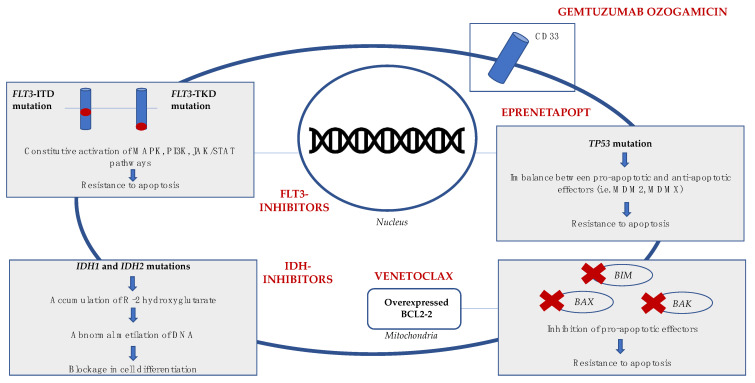
Selected cellular pathways targeted by new AML drugs.

**Figure 2 cancers-13-05075-f002:**
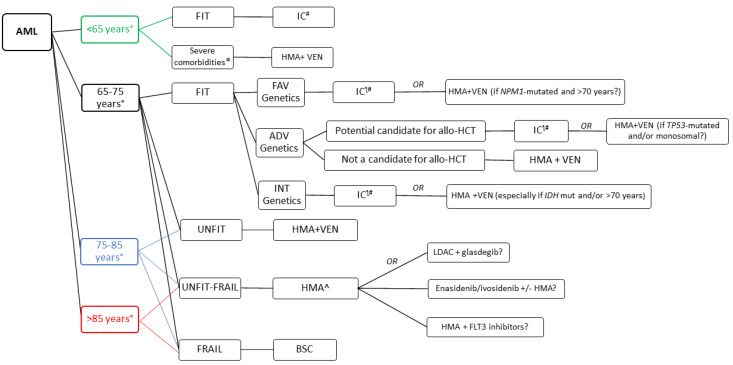
Treatment algorithm ^§^ of older AML patients that we follow in 2021. Abbreviations: Allo-HCT, allogeneic hematopoietic cell transplantation; ADV, adverse; BSC, best supportive care; FAV, favorable; HMA, hypomethylating agents; IC, intensive chemotherapy; INT, intermediate; LDAC, low-dose cytarabine; VEN, venetoclax. ^§^ The discussion regarding the type of consolidation or maintenance therapy is not included. ° Age cut-offs are somewhat arbitrary, and they are used to simplify the algorithm. They are never absolute. * In cases of moderate comorbidities, IC is usually preferred in this age group. In cases of significant cardiac comorbidity, a regimen without anthracycline can be considered (e.g., FLAG). ^#^ Intensive chemotherapy according to the disease profile (i.e., 7+3 and GO, 7+3 and midostaurin, CPX-351, FLAG-ida, 7+3). ^ƪ^ Above 70 years of age, dose reductions are usually considered, such as 2+5 with midostaurin and a 4-day regimen of FLAI. CPX-351 is administered at full dose when employed. ^ Especially when features predicting a good chance of response are present, such as non-proliferative disease.

**Table 2 cancers-13-05075-t002:** Main studies analyzing the outcomes of older patients with AML undergoing allogeneic HCT.

Reference	# Patients Diagnosis	Age	Allo-HCT	Graft Source	Conditioning	GVHD Acute II-IV	GVHD Chronic	LFS	OS	NRM
Bertz, JCO 2003 [101]	19 AML	60–70	MSD, *n* = 7; MUD, *n* = 12	PBSC	RIC	59%	65%	1 y 68%	1 y 61%	1 y 22%
Wong, Blood 2003 [102]	AML/MDS, *n* = 20 CML, *n* = 9	≥55	MUD	BM	RIC	41%	63%	1 y 37%	1 y 44%	1 y 55%
Yanada, BMT 2004 [103]	91 AML	≥50	MSD 80%, Alternative 20%	BM 74%, PBSC 21%	-	-	-	1 y 34%	1 y 35%	100 d 16%
Gupta, BBMT 2005 [104]	24 AML/MDS	≥60	MSD	PBSC	RIC	45%	74%	2 y 44%	2 y 52%	2 y 25%
Spyridonidis, Blood 2005 [105]	34 AML/MDS	≥60	MUD	BM 12%, PBSC 88%	RIC	-	-	2 y 53%	2 y 62%	2 y 20%
McClune, JCO 2010 [106]	63 AML/MDS	≥65	MSD 51%, MUD 49%	PBSC 97%	RIC	33%	53%	2 y 34%	2 y 36%	2 y 34%
Alatrash, BBMT 2011 [107]	AML, *n* = 63; MDS, *n* = 16	≥55	MSD 52%, MUD 48%	BM 48%, PBSC 52%	MAC, (Bu-Flu)	40%	43%	2 y 44%	2 y 46%	100 d 6%, 1 y 19%
De Latour, BMT 2015 [108]	714 AML	≥55	MSD, *n* = 404; MUD, *n* = 310	PBSC	RIC	MSD: 17%, MUD: 25%	MSD: 46%, MUD: 47%	3 y MSD:46%, MUD: 47%	3 y MSD: 49%, MUD: 49%	3 y MSD:17%, MUD: 23%
Kasamon, JCO 2015 [109]	271 AML/MDS	50–75	haplos	BM	RIC	33%	10%	3 y 40%	3 y 49%	1 y 12%
Haen, Blood Cancer J 2016 [110]	56 AML	≥70	MSD, *n* = 7; MUD, *n* = 49	PBSC, *n* = 55; BM, *n* = 1	MAC, RIC	16%	32%	3 y 43%	3 y 42%	2-y18%
Pohlen, BMT 2016 [111]	187 AML/MDS	≥60	MSD, MUD	PBSC 94%, BM 4%	RIC	-	-	3 y 32%	3 y 35%	1 y 37%
Bertz, Leukemia 2016 [112]	250 AML/MDS	≥60	MSD, *n* = 64; MUD, *n* = 186	PBSC	RIC	23%	37%	2 y 40%	2 y 48%	2 y 29%
Slade, BBMT 2017 [113]	95 AML/MDS	55–65 (*), ≥65 (#), ≥65 (§)	Haplos (*), Haplos (#), MUD (§)	PBSC	MAC, RIC	(*) 34%, (#) 14%, (§) 27%	(*) 35%, (#) 9%, (§) -	(*) -, (#) -, (§) -	(*) 2 y 34%, (#) 2 y 15%, (§) 2 y 24%	(*) 2 y 39% (#) 2 y 32% (§) 2 y 55%
Ringden, BBMT 2019 [114]	AML, *n* = 713; *n* = 16,161	≥70 (*), 50–69 (#)	MSD, MUD	PBSC, BM	RIC, MAC	(*) 23%, (#) 25%	(*) 43%, (#) 41%	(*)2 y 33%, (#) 2 y 44%	(*) 2 y 38%, (#) 2 y 50%	(*) 2 y 34% (#) 2 y 24%

Abbreviations. LFS, leukemia-free survival; OS, overall survival; NRM, non-relapse mortality; AML, acute myeloid leukemia; MDS, myelodysplastic syndrome; CML, chronic myeloid leukemia; MSD, matched sibling donor; MUD, matched unrelated donor; PBSC, peripheral blood stem cell; BM, bone marrow; RIC, reduced-intensity conditioning; MAC, myeloablative regimen; Haplos, haploidentical donor. *, # and § which are present in the last lines of the table, are used to refer to the different age groups evaluated in the study and are subsequently reported in the same line of the table to corresponding response outcomes.

**Table 4 cancers-13-05075-t004:** Recently approved drugs for older AML patients.

AML Subtype	Drug	Approval	Indication	Regimen	Phase Trial	cCR (%)	Median OS (months)
*FLT3*-mut	Midostaurin	FDA, EMA	ND AML	With 7+3 (DNR)	III	59	(51% at 4 years)
	Gilteritinib	FDA, EMA	R/R AML	Monotherapy	III	34	9.3
*IDH1*-mut	Ivosidenib	FDA FDA	ND AML in unfit R/R AML	Monotherapy Monotherapy	I I	42 (ND) 34 (R/R)	12.6 (ND) 8.8 (R/R)
*IDH2*-mut	Enasidenib	FDA	R/R AML	Monotherapy	I/II	20	8.8
CD33+ ELN fav/int	Gemtuzumab ozogamicin	FDA, EMA FDA FDA	ND AML R/R AML ND AML in unfit	With 7+3 (DNR) With 7+3 (DNR) Monotherapy	III III III	81 81 /	27.5 27.5 4.9
t(AML), AML-MRC	CPX-351	FDA, EMA	ND AML	Monotherapy	III	48	9.6
All	Venetoclax	FDA, EMA	ND AML in unfit	With HMA Or LDAC	III	66 (HMA) 48 (LDAC)	14.6 (HMA) 8.4 (LDAC)
All	Glasdegib	FDA, EMA	ND AML in unfit	With LDAC	II	17	8.8

Abbreviations. cCR, composite CR; OS, overall survival; DNR, daunorubicin; ND AML, newly diagnosed AML; R/R AML, relapsed/refractory AML; t(AML), therapy-related AML; AML-MRC, AML with myelodysplasia-related changes; FDA, Food and Drug Administration; EMA, European Medicines Agency; HMA, hypomethylating agents; LDAC, low-dose cytarabine.

**Table 5 cancers-13-05075-t005:** Genetic and clinical factors considered when choosing between hypomethylating agents alone, hypomethylating agents in combination with venetoclax and intensive chemotherapy.

Factors	Favor IC	Comment	Favor HMA+VEN	Comment	Favor HMA Alone	Comment
Age	<70 years	Globally inferior results above 70 years			>85 years	Toxicity concerns
WBC	Leukocytosis	Rapid disease control			Leukopenia	Poor disease control in proliferative disease
Blasts					BM blast <30%, few circulating blasts	Favorable outcomes in oligoblastic AML; circulating blasts associated with lower response rate
Genes						
*NPM1*	Mutated	Globally good results, although inferior in older adults	Mutated	High response rate, long duration of response in most reports	Wild type	Short duration of response in *NPM1*-mutated AML
*IDH2*			Mutated	High response rate in most reports, long duration of response		
*IDH1*			Mutated	High response rate in some reports		
*FLT3*-ITD	Mutated	Improved results with IC + FLT3 inhibitors			Wild type	Poor response and short survival with *FLT3* aberrations
*TP53*	Wild type	Low CR rates and poor survival in *TP53*-mutated cases	Mutated	Relatively high response rate (around 50%), but short duration of response	Mutated	
Secondary AML-like gene	Wild type	Poor outcomes if mutated; probable better results with CPX-351				
Signaling genes	Mutated	Promising role of GO	Wild type	Frequent association of these mutations with relapse and treatment resistance	Wild type	
Cytogenetics	Favorable	Very high CR rate, prolonged OS	Poor	Relatively high response rate	Poor	
Phenotype			Non-monocytic	Treatment resistance, lower response rates in monocytic AML		
AML type	De novo	However, better results with CPX-351 in secondary AML	Secondary AML	No significant impact on response	Secondary AML	

Abbreviations. BM, bone marrow; CR, complete remission; IC, intensive chemotherapy; HMA, hypomethylating agents; VEN, venetoclax; WBC, white blood cells.

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
