# Peer review of "Evolving Therapeutic Approaches for Older Patients with Acute Myeloid Leukemia in 2021"

_cancers, 2021, doi:10.3390/cancers13205075_

Round 1
Reviewer 1 Report
Dear Authors,
This a well written comprehensive review on the current/evolving therapeutic approaches for acute myeloid leukemia (AML) for patients out there especially in older patients.
- Authors extensively covered prediction models which are based on different score variables for mortality/survival in Table 1.
- The flow of the paper is nice. Authors started off with “intense therapeutic approaches” for “fit” patients that include induction chemotherapy, Ab-drug conjugates, Targeting agents followed by consolidation therapy, Autologous/allogenic stem cell transplantation, maintainance therapy that include hypomethylating agents (HMA’s).
- This is followed by “less intensive approaches” for patients that are “not fit” including BCL2 inhibitors: venetoclax based combinations, hedgehog pathway inhibitor: glasdegib, FLT3 inhibitor: gilteritinib, IDH inhibitors: ivosidenib and enasidenib, and anti-CD33 monoclonal antibody ; gemtuzumab ozogamicin. All these therapeutic approaches are covered in detail including their efficacy and toxicity. Authors have also assessed the risks and benefits in choosing the particular treatment choice depending on patients age, performance status etc.
- Authors did a fantastic job in consolidating treatment options as shown in Figure 1. This is where readers or physicians can really take advantage of when it comes to understanding/decision making of the treatment choice for patients with AML especially in older patients.
- The future looks bright for even better therapeutic approaches as more and more monoclonal, bispecific antibodies (more specific and less toxic than traditional small molecules) being tested in different preclinical/clinical trials.
- Authors did a good job in having references for most of their statements which is crucial for review paper.
- The authors would need to look for typos/spelling checks throughout the document. For example, Page 2, line 89: please change it to “care” instead of car.
Author Response
Point 1: The authors would need to look for typos/spelling checks throughout the document. For example, Page 2, line 89: please change it to “care” instead of car.
Response 1: We thank the reviewer for his overall appreciation of our manuscript. We looked at typos/spelling checks throughout the document providing errors correction, as suggested.
Reviewer 2 Report
This a a well organized, updated and comprehensive review of treatment options for acute myeloid leukemia (AML) in the elderly population. A few comments for improvement are listed below.
- A brief, though comprehensive, paragraph of the genetics of AML in the elderly population would be beneficial to the reader for a better understanding of targeted treatment. A dedicated figure illustrating the main draggable molecular pathways would also be useful.
- Quality of Life (QoL) is very important when dealing with AML in the elderly population. Data on patient reported outcomes (PROs) should also be included. I suggest to expand the section on QoL and PROs to a certain extent. In particular, the paragraph on Health related QoL may also include a mention to the need of new models for generating health utilities for economic research in elderly AML. In this respect, a recently proposed model for myelodysplastic syndromes (MDS) might provide useful insights also for AML in the elderly population, that are frequently related to a previous MDS condition (see Gamper et al, J Clinical Epidemiol 137:31, 2021).
- Evolution of AML from myeloproliferative neoplasms should be covered in more detail, also referring to important new data (Marchetti et al. Am J Hematology 95:295, 2020)
Author Response
Point 1: A brief, though comprehensive, paragraph of the genetics of AML in the elderly population would be beneficial to the reader for a better understanding of targeted treatment. A dedicated figure illustrating the main draggable molecular pathways would also be useful.
Response 1: We thank the reviewer for his comments. Following the reviewer’s suggestions, we better explained the genetics of AML in the elderly and its implications (Paragraph 4) and we represented the main draggable molecular pathways in figure 1 of the revised version.
Point 2: Quality of Life (QoL) is very important when dealing with AML in the elderly population. Data on patient reported outcomes (PROs) should also be included. I suggest to expand the section on QoL and PROs to a certain extent. In particular, the paragraph on Health related QoL may also include a mention to the need of new models for generating health utilities for economic research in elderly AML. In this respect, a recently proposed model for myelodysplastic syndromes (MDS) might provide useful insights also for AML in the elderly population, that are frequently related to a previous MDS condition (see Gamper et al, J Clinical Epidemiol 137:31, 2021).
Response 2: We thank the reviewer for this comment. We have expanded the paragraph on HRQoL (paragraph 6.2) addressing the role of patient reported outcomes (PROs) in AML older patients (Peipert JD et al. J Geriatr Oncol. 2021). We also briefly discussed the need of HRQoL models for health economic research (Rowen et al, Pharmacoeconomics. 2017), referring to the recently proposed model for MDS as suggested by the reviewer.
Point 3: Evolution of AML from myeloproliferative neoplasms should be covered in more detail, also referring to important new data (Marchetti et al. Am J Hematology 95:295, 2020)
Response 3: We agree that the therapeutic approach for post-MPN AML could have been further discussed. Considering the dismal prognosis of this group of AML patients with standard therapy, we decided to address this topic referring to future therapeutic perspectives and we added paragraph 4.5.6 on JAK2 inhibitors combinations currently under investigations.
Reviewer 3 Report
The authors have done a commendable job in this exhaustive review of the treatment of acute myeloid leukemia in older adults. Figure 1 is the highlight of this manuscript providing a framework for the current treatment paradigm. I suggest the author include a table for current approved targeted agents (FDA, EDA) with response rates and phase of the study used for approval. Also, suggest a table for drugs being studied in the future perspectives sections.
Minor comment:
- Line 322. Replace word "addiction" with "addition"
Author Response
Point 1: I suggest the author include a table for current approved targeted agents (FDA, EDA) with response rates and phase of the study used for approval. Also, suggest a table for drugs being studied in the future perspectives sections.
Response 1: We thank the reviewer for his positive feedback. We have followed the reviewer’s suggestions and added table 3 (for selected future therapeutic perspectives) and table 4 (for current approved targeted agents).
Point 2: Minor comment: Line 322. Replace word "addiction" with "addition"
Response 2: We provided error correction
Reviewer 4 Report
In this review, Urbino et al detail the approaches used in AML touching older patients. The manuscript is clear with numerous references. Review about treatment of AML in older patients appeasr relevant as this population is numerous and evolution in treatment strategies is fast.
A minor correction The authors must detailed that the review excludes t(15;17)
Author Response
Point 1: A minor correction The authors must detailed that the review excludes t(15;17).
Response 1: We thank the reviewer for his overall appreciation of our manuscript. We specified that acute promyelocytic leukemia is not addressed in the review (page 2, line 73).
Reviewer 5 Report
With this manuscript the group of Dr Cerrano provides a well written and exhaustive review exploring the eligibility criteria in different types of patients (FIT, UNIFIT and FRAIL) regarding several types of chemotherapy treatment, customizing it according to patient’s clinical conditions.
An interesting look is also placed on the prospect of future therapies emphasizing, towards the end of the review, the importance of the patient’s quality of life.
Minor comments:
- The authors should proofread the review, as there are some minor grammar errors. For instance line 91 “care” and not “car” or line 836 “tools” and not “tolls”
- Please check that acronyms are specified when first appearing in the paper. E.g. line 501 “hypomethylating agents” (HMA) is already specified on line 403. Same aspect for “Overall Response Rate” (ORR) that first appears on line 509 or “QoL” on line 179 and not on line 842. Whereas “ADL” on line 177 is never used.
- Authors should check that genes and proteins symbols are respectively in italics and in capital letters in chapters 4.5.1 and 4.5.4.
- Chemotherapy treatment with 3 days of Anthracycline and 7 days of Cytarabine is commonly referred as “7+3” regimens (as in ref [20]) so please replace, from line 257, “3+7” with “7+3” and also move the ref [20] to the end of the sentence on line 259.
- I would suggest, if possible, to add a table summarizing the less-intensive approaches of chapter 4.
Author Response
Point 1: The authors should proofread the review, as there are some minor grammar errors. For instance line 91 “care” and not “car” or line 836 “tools” and not “tolls”
Response 1: We thank the reviewer for his remarks. We looked at minor grammar errors throughout the document providing correction.
Point 2: Please check that acronyms are specified when first appearing in the paper. E.g. line 501 “hypomethylating agents” (HMA) is already specified on line 403. Same aspect for “Overall Response Rate” (ORR) that first appears on line 509 or “QoL” on line 179 and not on line 842. Whereas “ADL” on line 177 is never used.
Point 3: Authors should check that genes and proteins symbols are respectively in italics and in capital letters in chapters 4.5.1 and 4.5.4.
Point 4: Chemotherapy treatment with 3 days of Anthracycline and 7 days of Cytarabine is commonly referred as “7+3” regimens (as in ref [20]) so please replace, from line 257, “3+7” with “7+3” and also move the ref [20] to the end of the sentence on line 259.
Response 2-3-4: We apologize for the reported inaccuracies, and we have now corrected them.
Point 5: I would suggest, if possible, to add a table summarizing the less-intensive approaches of chapter 4.
Response 5: We thank the reviewer for the suggestion. We have now added table 4, which describes less-intensive options and the other recently approved targeted agents. We think that a common table could be useful to summarize the entire range of new drugs currently available for older AML patients, also considering space constraints.